# Enlisting the *Ixodes scapularis* Embryonic ISE6 Cell Line to Investigate the Neuronal Basis of Tick—Pathogen Interactions

**DOI:** 10.3390/pathogens10010070

**Published:** 2021-01-14

**Authors:** Lourdes Mateos-Hernández, Natália Pipová, Eléonore Allain, Céline Henry, Clotilde Rouxel, Anne-Claire Lagrée, Nadia Haddad, Henri-Jean Boulouis, James J. Valdés, Pilar Alberdi, José de la Fuente, Alejandro Cabezas-Cruz, Ladislav Šimo

**Affiliations:** 1UMR BIPAR, Laboratoire de Santé Animale, ANSES, INRAE, Ecole Nationale Vétérinaire d’Alfort, Paris-Est Sup, 94700 Maisons-Alfort, France; lourdes.mateos@vet-alfort.fr (L.M.-H.); eleonore.allain@hotmail.fr (E.A.); clotilde.rouxel@vet-alfort.fr (C.R.); anne-claire.lagree@vet-alfort.fr (A.-C.L.); nadia.haddad@vet-alfort.fr (N.H.); henri-jean.boulouis@vet-alfort.fr (H.-J.B.); 2Faculty of Science, Pavol Jozef Šafarik University in Košice, 04180 Košice, Slovakia; kokosova.natalia@gmail.com; 3AgroParisTech, Micalis Institute, Université Paris-Saclay, PAPPSO, INRAE, 78350 Jouy-en-Josas, France; celine.henry@inrae.fr; 4Institute of Parasitology, Biology Centre of the Czech Academy of Sciences, Branisovska 31, 37005 Ceske Budejovice, Czech Republic; valdjj@gmail.com; 5Department of Virology, Veterinary Research Institute, Hudcova 70, 62100 Brno, Czech Republic; 6SaBio Instituto de Investigación en Recursos Cinegéticos IREC-CSIC-UCLM-JCCM, Ronda de Toledo s/n, 13005 Ciudad Real, Spain; maria.alberdi@uclm.es (P.A.); josedejesus.fuente@uclm.es (J.d.l.F.); 7Neuroplasticity and Neurodegeneration Group, Regional Centre for Biomedical Research (CRIB), Ciu-dad Real Medical School, University of Castilla-La Mancha, 13071 Ciudad Real, Spain; 8Center for Veterinary Health Sciences, Department of Veterinary Pathobiology, Oklahoma State University, Stillwater, OK 74078, USA

**Keywords:** *Ixodes scapularis* ISE6 cell line, neuropeptides, *Anaplasma phagocytophilum*, synganglion

## Abstract

Neuropeptides are small signaling molecules expressed in the tick central nervous system, i.e., the synganglion. The neuronal-like *Ixodes scapularis* embryonic cell line, ISE6, is an effective tool frequently used for examining tick–pathogen interactions. We detected 37 neuropeptide transcripts in the *I. scapularis* ISE6 cell line using in silico methods, and six of these neuropeptide genes were used for experimental validation. Among these six neuropeptide genes, the tachykinin-related peptide (TRP) of ISE6 cells varied in transcript expression depending on the infection strain of the tick-borne pathogen, *Anaplasma phagocytophilum*. The immunocytochemistry of TRP revealed cytoplasmic expression in a prominent ISE6 cell subpopulation. The presence of TRP was also confirmed in *A. phagocytophilum*-infected ISE6 cells. The in situ hybridization and immunohistochemistry of TRP of *I. scapularis* synganglion revealed expression in distinct neuronal cells. In addition, TRP immunoreaction was detected in axons exiting the synganglion via peripheral nerves as well as in hemal nerve-associated lateral segmental organs. The characterization of a complete *Ixodes* neuropeptidome in ISE6 cells may serve as an effective in vitro tool to study how tick-borne pathogens interact with synganglion components that are vital to tick physiology. Therefore, our current study is a potential stepping stone for in vivo experiments to further examine the neuronal basis of tick–pathogen interactions.

## 1. Introduction

The North American black-legged deer tick *Ixodes scapularis* and the European castor-bean tick *Ixodes ricinus* are both medically important arthropod vectors. These two allopatric tick species are well recognized for transmitting a wide spectra of bacterial, viral, and protozoan pathogens [1,2]. Both tick species infect hosts with the bacterium *Anaplasma phagocytophilum*, which is an emerging tick-borne pathogen (TBP) that causes granulocytic anaplasmosis and tick-borne fever [3]. Therefore, a tick control strategy is needed for *A. phagocytophilum*, since the infection rate for humans and animals increase yearly in the United States, Europe, Africa, and Asia [4]. However, developing effective tick control strategies to prevent *A. phagocytophilum* transmission requires a better understanding of specific interactions between this pathogen and the tick [5]. To address this limitation, researchers may utilize the *I. scapularis* embryo-derived cell line, ISE6, as an in vitro analytical tool, since ISE6 cells are commonly used in propagating various TBPs [6,7,8,9,10]. The ISE6 cell line is successful in isolating and analyzing tick-borne bacteria (i.e., *Rickettsia*, *Ehrlichia*, *Borrelia,* and *Anaplasma*) [8,10,11,12,13,14], as well as propagating arboviruses and vector-borne flaviviruses (i.e., Semiliki Forest, Hazara, or the Langat viruses) [15,16].

The *I. scapularis* ISE6 cell line was established almost three decades ago [17], but its tissue origin, or biological background, remains ambiguous. However, proteomics and transcriptomics analyses of ISE6 cells reveal a mixture of neuronal and immune response-associated markers [18,19]. Although the potential neuronal function of ISE6 cells remains untested, the expression of multiple neuron-associated proteins is confirmed [18]. For instance, Slit molecules promote neuronal axon guidance, cell mobility and permeability, and viral infection. The Roundabout protein 2, the transmembrane receptor for Slit, protects ISE6 cells from *A. phagocytophilum* infection [20]. Morphologically, in vivo exposure of ISE6 cells to the tick hemocoel triggers a neuron-like phenotype [18]. Immune activators carried by *A. phagocytophilum* also cause an enhanced pathogenic resistance in ISE6 cells via a typical immune cell memory-like property known as immune priming [21,22]. Specifically, ISE6 cells primed with lipids 1-palmitoyl-2-oleoyl-sn-glycero-3-phosphoglycerol and 1-palmitoyl-2-oleoyl diacylglycerol activate the immune deficiency pathway to reduce the *A. phagocytophilum* burden compared with naïve cells [22]. Therefore, the ISE6 cell population is composed of neuron-like and hemocyte-like subpopulations [18,19,21], although the neuron-like phenotype is predominant [18].

*A. phagocytophilum* infects vertebrate granulocytes, whereas tick infection sites include the midgut, hemocytes, and salivary glands [3]. Although pathogen infection may increase tick fitness and survival, rather than cause harm [23], several studies using “omics” technologies reveal that *A. phagocytophilum* infection induces both transcriptional reprograming and protein-level changes in vertebrate and arthropod host cells [24,25,26,27]. The genetic and molecular alterations in ticks coincide with the multi-host adaptation of *A. phagocytophilum* [28,29]. The main cellular components and processes affecting ticks by *A. phagocytophilum* infection are the cytoskeleton, immunity, epigenetics, apoptosis, and metabolism [25,26,27,30,31]. Recent studies also suggest that *A. phagocytophilum* modulates complex behavioral responses in infected ticks [32,33,34,35]. Although the capacity of *A. phagocytophilum* to manipulate cellular processes is well established, the interaction of specific signaling molecules within the tick central nervous system, i.e., the synganglion, is overlooked. The first report regarding a bacterial pathogen infecting tick synganglion was from the *Ehrlichia muris*-like agent, a closely related bacterium to *A. phagocytophilum* [36]. The cortical zone of *I. scapularis* synganglion was heavily infected when fed with *Ehrlichia*-infected hamsters that indicated pathogen crossing of tick synganglion barriers [36].

Neuropeptides are expressed predominantly in multiple tick synganglion neurons [37,38]. These small peptide molecules, along with their cognate G protein-coupled receptors, are important for invertebrate physiology, since their signaling pathways regulate key processes in reproduction, development, growth, and behavior [39,40]. Neuropeptidergic cells synthesize the neuropeptide precursor, called prepropeptide that is usually composed of multiple copies of the same (or similar) neuropeptides. The prepropeptide signal peptide drives transportation for cleavage of the prepropeptide into smaller, mature neuropeptides. Then, these mature neuropeptides are packed into secretory vesicles that are eventually transported to axon terminals for exocytosis [41]. The number of neuropeptide-encoding genes among invertebrates is more or less the same; however, their specific biological functions may differ among particular taxa [42]. Theoretically, the *I. scapularis* genome consists of approximately 100 neuropeptides encoded by at least 34 genes that correspond to invertebrate neuropeptide orthologs [37,43]. Tick neuropeptides are identified with immunohistochemistry, proteomics, and genomics [37,38,44,45], but their function is almost exclusively implicated by discovering their transport pathways and/or the localization of their cognate receptors in target organ(s) [37,46,47]. Tachykinin-related peptides (TRP) constitute a large family of pleiotropic neuropeptides found across bilaterians [48]. On the one hand, vertebrate TRPs play important roles in pain, inflammation, sensory processes, immune systems, gut function, or hormonal regulation [48]. On the other hand, invertebrate TRPs are known for regulating the central nervous system and gut function, while olfactory processing, locomotion, food seeking, vasodilatation, nociception, and metabolic stress are also described [48].

For our current study, we confirmed the presence of an entire *Ixodes* neuropeptidome in ISE6 cells that led to several questions. Are neuropeptide-expressing ISE6 cells peptidergic neurons? Are *Ixodes* TRPs expressed in vivo (i.e., *I. scapularis* synganglion)? How are *Ixodes* TRPs respectively distributed in vitro versus in vivo? Given that *A. phagocytophilum* induces transcriptional reprograming in ticks [24,25,26], will infection affect *Ixodes trp* gene expression? The cumulative answers to these questions propose that ISE6 cells may serve as an effective in vitro tool for studying the nature of tick neuropeptidergic cells and their interactions with various TBPs.

## 2. Results

### 2.1. Neuropeptidome of ISE6 Cells

Extensive BLAST searches in the ISE6 genome predict 38 distinct genome-based neuropeptide genes and their representative genomic scaffolds (Table 1). The presence of 37 neuropeptide transcripts was also confirmed in the ISE6 Sequence Read Archive (SRA) databases (BioProject: PRJNA239331). The transcript for natalisin was the only neuropeptide not detected in our in silico searches (Table 1). We also failed to experimentally amplify the natalisin transcript using three different sets of primers (Appendix A). Then, six of the SRA-confirmed neuropeptides (i.e., sulfakinin, kinin, CCHamide, short neuropeptide F, and FGLamide-related allatostatin) were selected for qRT-PCR validation using RNA extracted from ISE6 cells (Appendix A).

### 2.2. Structure of the Gene-Encoding Ixodes TRP

The ISE6 genome BLAST yielded a predicted transcript (VectorBase accession number ISCI008383) encoding a putative TRP. The *I. ricinus* TRP transcript (GenBank accession number MW082607) was molecularly identified in this study. Nucleotide and protein alignments of the *I. scapularis* TRP (ISCI008383) with *I. scapularis* TRP EST (EL516783) and *I. ricinus* TRP (MW082607) reveal an incorrect computational prediction of ISCI008383. Specifically, the 5′-end of ISCI008383 is incomplete with an incorrect translated putative signal peptide. The 3′-end reading frame is also shifted, causing an improper conceptual TRP translation (Appendix A). A BLAST using the *I. scapularis* EST (EL516783) encoding TRP, against the ISE6 genome confirms a relationship to the PKSA02005591.1 scaffold, resulting in a better representation of the *I. scapularis* TRP genomic organization (Figure 1A). In ISE6 cells genome, the *I. scapularis trp* gene (Figure 1A) is composed of four exons (≈140, 87, 328, and 170 bp) interrupted by three introns (189857, 5018, and 1862 bp). The *I. scapularis trp* ORF is 492 bp, spanning exons 2–4 (Figure 1A).

The translated TRP ORF of *I. scapularis* (169 residues) and *I. ricinus* (172 residues) share 96.5% amino acid identity (Figure 1B). The TRP prepropeptide contains dibasic cleavage sites for three putative TRPs (plus one repeat) for both tick species that are characterized by a general conserved carboxy-terminus motif, F-x_1_-G/A-x_3_-Ramide (Figure 1B,C). This TRP motif is typical for other tick species such as *Rhipicephalus sanguineus, Rhipicephalus microplus*, and *Dermacentor silvarum*, as well as the parasitic mite *Varroa destructor* [49] and the fruit fly *Drosophila melanogaster* TRPs (Figure 1C). *I. scapularis*, *I. ricinus*, *R. sanguineus*, *R. microplus* and *D. silvarum* share identical TRP1 and TRP2 amino acid sequence (Figure 1C).

### 2.3. Expression of ISE6 TRP in Response to A. phagocytophilum Infection

Compared to uninfected ISE6 cells (control), *A. phagocytophilum* infection causes disparate *trp* transcript levels that are strain-dependent (Figure 2A). Specifically, *trp* levels significantly decrease 0.5-fold in ISE6 cells infected with human strain NY18, but they significantly increase 2.5-fold with bovine strain BV49. No significant changes in *trp* transcript levels were detected in ISE6 cells infected with *A. phagocytophilum* ovine strain NV2Os (Figure 2A). Therefore, *A. phagocytophilum* NV2Os was selected to infect ISE6 cells for subsequent immunocytochemistry (ICC) detection of TRP, since *trp* transcript levels were stable (Figure 2B–G). Immunochemical analyses were facilitated by an antibody against the *D. melanogaster* neuropeptide natalisin (DromeNTL4) [50], (also see Section 4) that is a sister group of TRP. The DromeNTL4 C-terminal sequence (FPATRamide) is highly similar to the *Ixodes* TRP3 (FVATRamide) (Figure 1C). Therefore, the term TRP-like immunoreaction (TRP-like IR) is used hereafter.

The ICC revealed cytoplasmic TRP-like IR in the majority of uninfected ISE6 cells (Figure 2B,C). Although processes such as axon-like filopodia were apparent in a predominant cell subpopulation of both uninfected and infected ISE6 cells, these filopodia were absent of any TRP-like IR (Figure 2B–G). The levels of NV2Os used (60% and 80%) are considered a high infection status that may cause a decrease in cell population—as qualified by cells infected with 80% NV2Os compared to uninfected and 60% infection (Figure 2D–G). As expected by the stable *trp* transcript levels, and comparable to uninfected cells (Figure 2A), cytoplasmic TRP-like IR was also observed in ISE6 cells from both NV2Os infection levels (Figure 2D–G). However, there are more compact clusters of TRP-like IR in uninfected and 60% NV2Os-infected cells than at 80% infection (Figure 2B–G). Negative control in uninfected and infected cells did not reveal any TRP-like IR (Figure 2H–M).

### 2.4. Expression of TRP in I. scapularis Synganglion

The in situ hybridization (ISH) determined *trp* transcript distribution in specific neurons of the *I. scapularis* synganglion (Figure 3A,B). Each positively labeled neuron is represented as the first two letters of a specific synganglion lobe: prothocerebral (Pc), pedal ganglion 1–4 (Pd1–4), opisthosomal ganglion (Os). The letters that follow refer to the anatomical location of the neuron: dorsal (D), ventral (V), anterior (A), posterior (P), medial (M), or lateral (L). The Pc *trp* transcripts are expressed in three pairs of small PcAM neurons, two pairs of small PcDL_1,2_ neurons, and one pair of small PcDM neurons. The *trp* signal in the Pd1 was detected in a prominent pair of large Pd_1_DL neurons surrounded by two pairs of smaller Pd_1_DL_1,2_ and Pd_1_VL_1,2_ neurons. The *trp* signal was also detected in the Pd3 by a small pair of neurons in the Pd_3_DM and the Pd_3_VL_1,2_. Strong *trp* transcript signals identified on the dorsal side of the Os were expressed by two pairs of OsDM and OsDM_1_ neurons, while signals from the ventral side were detected by three pairs of OsVM_1–3_ neurons.

The immunohistochemistry (IHC) confirms the ISH reaction in neurons of the PcAM, Pd_1_DL, Pd_3_VL_1_, OsDM, and OsDM_1_ (Figure 3C,D). A strong TRP-like IR was detected in the axons exiting the prominent Pd_1_DL neurons. These axons run from the Pd_1_DL neuronal bodies toward the esophagus, turning laterally about the level of the second pedal ganglion and arborize into a rich axonal network on the synganglion surface (Figure 3C,I). The TRP-like axons of the lateral nerves and their three hemal branches originate from unidentified neurons of the Os (Figure 3E–I). The TRP-like axons in hemal nerves 1 and 2 run alongside the lateral segmental organs (LSO) (Figure 3E–G). TRP-like IR was scattered in all four intrinsic LSO cells. The most prominent reactions in two LSO cells are in contact zones with the TRP-like axons (Figure 3F,G). TRP-like axons exiting the synganglion were also detected in all four pairs of nerves exiting the Os (Figure 3H,I). Scattered TRP-like IR was detected in axons terminating on the dorsal surface of periganglionic sheath. The origin of this axons are likely Pd_1_DL neurons. 

## 3. Discussion

The relationships between ticks and TBPs are complex, and understanding their molecular determinants is crucial for developing effective control strategies. Here, we introduce the existence of neuropeptide transcripts in ISE6 cells and present their neuropeptidergic features. These findings are supported by previous studies indicating that ISE6 cells are predominantly neuron-like [18].

The in silico prediction of neuropeptide genes from the ISE6 cell genome [52] reveals the same set of genes identified in the *I. scapularis* genome [37,43], but previous proteomic approaches failed to identify mature neuropeptides in ISE6 cells [18]. In the current study, we confirmed the expression of a complete *Ixodes* neuropeptidome in ISE6 cells. Considering that neuropeptides are regulators of all tick physiological processes and pathogens modulate tick physiology [32], our study suggests that ISE6 cells are an effective in vitro archetype for investigating TBP interactions with vital elements (i.e., neuropeptides) of the tick synganglion. Therefore, investigating tick–pathogen interactions by enlisting parallel, yet similar, cell-types (i.e., ISE6 cells and tick synganglion) may contribute to advancing tick control strategies to prevent TBP transmission. Attempts have been made to exploit components of the tick nervous system for control measures, but these studies did not achieve any information for pathogen infection or transmission [53,54,55].

Our ICC analyses detect distinct TRP-like IR in ISE6 cells, indicating the effective translation process of *trp* transcripts. It is not known whether the TRP-like IR in ISE6 cells is specific to neuropeptide epitopes within the prepropeptide, propeptide, or mature neuropeptides. However, evidence shows that mature neuropeptides are transported via the axons to their terminals [41], while pre/propeptides, theoretically, are proximate to the cell cytoplasm. Our study shows that TRP-like IR was strictly localized within the cytoplasm of ISE6 cells, while no staining was expressed in the axon-like filopoda. The ISE6 cells develop axon-like projections under unknown factors in the hemocoel of partially fed ticks [18]. Therefore, an interesting examination is if these unknown factors facilitate the axonal guidance of specific neuropeptidergic-type ISE6 cells. The subsequent immunodetection of additional neuropeptides may also elucidate the qualitative distribution of neuropeptidergic-type ISE6 cells.

The African species *Rhipicephalus appendiculatus* was the first tick experimentally used to confirm TRP-like IR in synganglion neurons [38]. The *R*. *appendiculatus* Pd_1_SG neurons are named Pd_1_DL neurons in *I. scapularis* where we localized, among other synganglion cells, *trp* transcripts using ISH. Our IHC approach confirms TRP-like IR in the majority of ISH synganglion stained cells, thereby highlighting specific *I. scapularis* TRP-producing neurons. The molecular characterization of the *I. ricinus* TRP encoding sequence also confirms an evolutionary relationship with the *I. scapularis* TRP previously identified [56] and verified here. The commonality in TRP-like expression for *R. appendiculatus* and *I. scapularis* synganglion, and the sequence identity of two out of three mature TRP between *I. scapularis*, *I. ricinus*, *R. microplus*, *R. sanguineus* and *D. silvarum*, suggests common attributes of TRP signaling in hard tick lineage.

In addition to the TRP-like IR in neuronal bodies, we identified several TRP-like IR axons exiting the *I. scapularis* synganglion, suggesting multiple visceral targets for TRP–axonal signaling. Localized TRP-like IR in LSOs, and their associated axons, supports previous findings that LSO cells are peptidergic and/or serve as neurohemal sites [37,38]. Furthermore, the arborization of TRP-like axon terminals on the paraganglionic sheath of the dorsal synganglion surface suggests a possible neurohemal site for discharging TRP to the hemolymph—thereby acting as neurohormones. Knowledge on TRP roles in ticks may help to localize their cognate receptors. The *Ixodes* genome possesses 10 putative TRP receptors (TRP-Rs) [43], whereas insects, at most, possess two TRP-Rs per species [57]. Since a TRP function in insects is diuresis [57], the increased number of *Ixodes* TRP-Rs may reflect the acute need for eliminating excess water and metabolic waste during tick feeding. However, whether these *Ixodes* TRP-Rs are functional and/or possess affinity to TRP ligands awaits experimental confirmation.

*A. phagocytophilum* induces tissue-specific transcriptional reprogramming, thereby affecting different cellular functions in infected tick [21,22,23,26,27,28]. *A. phagocytophilum* infection also alters tick physiology and behavior [32]. For example, *A. phagocytophilum*-infected ticks were more fitted to survive in cold temperatures [34] or desiccating conditions [35] compared to uninfected ticks. The differential regulation of heat shock protein transcripts (i.e., *hsp20* and *hsp70*) upon *A. phagocytophilum* infection in ticks was associated with an increase in questing activity [35]. Our present study shows that *trp* levels in ISE6 cells are differentially regulated in response to infection by different *A. phagocytophilum* strains with specificities for bovine (BV49), ovine (NV2Os), or human (NY18) hosts. Although the precise molecular mechanism(s) by which particular *A. phagocytophilum* strains interact with TRP is unknown, our data suggest that *A. phagocytophilum* modulates TRP transcript levels in a strain-specific manner. Invertebrate TRPs have multiple functions in the central nervous system and intestine [48], but the specific functions of many tick neuropeptides have yet to be determined. Therefore, investigations are necessary to conclude if *A. phagocytophilum* affects TRP expression in infected ticks and thus alter their physiology.

Future research will confirm if other neuropeptide transcripts differentially respond to the presence of *Anaplasma* to elucidate the complex cascade of physiological features potentially modified by this pathogen. We anticipate that identifying the crucial neuronal components in tick–pathogen interactions will present key targets for developing novel tick management strategies applicable to a broad spectrum of TBPs. Thereby, these novel strategies will reduce the negative impacts TBPs have on human and animal health.

## 4. Materials and Methods

### 4.1. In Silico Identification of Neuropeptides Genes in ISE6 Databases

We used 34 *I. scapularis* preproneuropeptide query sequences previously identified in the genome project [43]. The list of queries was enriched by previously identified *I. scapularis* neuropeptide sequences natalisin [49] and elevenin [58], and the insulin-like peptide 2, identified using *Drosophila subobscura* sequence (GenBank access. no. XP_034657457.1). In addition, an isoform b of *I. scapularis* crustacean hyperglycemic hormone-(CHH)-related ion transport peptide (CHH/ITP) was also included to the query list. Thus, 38 preproneuropeptides were BLAST against publicly available ISE6 cell line databases in NCBI (www.ncbi.nlm.nih.gov) and VectorBase (www.vectorbase.org) to identify a computed predicted transcript of homologous neuropeptides. To identify genomic scaffolds encoding neuropeptides, VectorBase databases were exclusively used for BLAST. To reveal the expressed neuropeptide transcripts in ISE6 cells, we provided homology BLAST searches of the ISE6 Sequence Read Archive (SRA) in BioProject PRJNA239331 that contains 33 experimental datasets from Illumina transcript reads.

### 4.2. Culture of ISE6 Cells

The ISE6 embryonic tick cell cultures were maintained according to Munderloh et al. [17]. Healthy, uninfected ISE6 cells were propagated in a flask of 25 cm^2^ with 5 mL of L15B300 medium. Infected cells were cultured in L15B300 medium supplemented with 0.1% NaHCO_3_ and 10 mM HEPES with an adjusted pH at 7.5. Both uninfected and infected ISE6 cells were maintained at 34 °C.

### 4.3. Neuropeptides Quantitative Real-Time PCR in ISE6 Cells

Total RNA was extracted from ISE6 cells by Trizol Reagent (Invitrogen, Carlsbad, CA, USA) and complementary DNA (cDNA) was obtained by reverse transcription using the High Capacity cDNA Reverse Transcription kit (Invitrogen, Carlsbad, CA, USA). The template cDNAs were analyzed for amplification using the SYBR Green Master Mix (Roche, Basel, Switzerland) on a LightCycler^®^ 480 thermocycler (Roche, Basel, Switzerland). The primers for quantitative RT-PCR (qRT-PCR) are listed in Appendix A. Then, relative transcript levels were calculated using the ΔΔCt ratio [59]. The ribosomal protein S4 (*rps4*) (GenBank accession number DQ066214) was used as a reference gene [60]. The statistical significance of normalized Ct values between groups was evaluated by Student’s *t*-test with unequal variance in the GraphPad 5 Prism program (GraphPad Software Inc., San Diego, CA, USA). Differences were considered significant when *p* < 0.05. Three technical and two biological replications were performed. 

### 4.4. Gene Cloning and Sequence Analyses

The *I. scapularis* EST sequence (EL516783) [44] was used to amplify the full open reading frame (ORF) of *I. ricinus* TRP. The forward and reverse primers used for amplification were 5′-AGTGATAAGCAAACCCGGTG-3′ and 5′-CACGGCTTGGGGAATCTTCT-3′, respectively. The predicted full-length ORF of *trp* was amplified by PCR using cDNA isolated from unfed *I. ricinus* adult synganglia. The PCR amplicon of *trp* ORF was inserted into the pGEM-T Easy vector (Promega) followed by the transformation of competent DH5α bacteria (prepared using the Mix & Go kit, Zymo Research). Plasmid DNA was purified using the Nucleospin Plasmid kit (Macherey-Nagel, Düren, Germany). Recombinant plasmids were commercially sequenced (Eurofins, Luxemburg). We used the Signal P3.0 server to predict the signal peptide of TRP precursor [61].

### 4.5. Pathogen Infection of ISE6 Cells

Either the *A. phagocytophilum* bovine strain BV49 [62], ovine strain NV2Os [63], or human strain NY18 [64] were propagated in ISE6 tick cells as described before [19]. *A. phagocytophilum* infection was propagated by transferring 1/10th of an infected ISE6 cell culture to a new flask of healthy cells once infection reached 70%. To determine the level of infection, 300 μL of media with 100 μL of suspended cells were mixed, and 60 μL of the mixture were concentrated on a slide using the Shandon Cytospin (Thermo Fisher Scientific, Kalamazoo, MI, USA). Subsequently, the Hemacolor^®^ staining kit (Merck, Darmstadt, Germany) provided observation of cells under an Olympus BX53 light microscope (Olympus, Hamburg, Germany).

### 4.6. Immunocytochemistry of TRP in Uninfected and A. phagocytophilum-Infected ISE6 Cells

The cells were cultured in tubes of 7 cm^2^ with 1 mL of L15B300 medium. Trac bottles containing internal glass slides (Dutscher, Brumath, France) were maintained at 34 °C as described in the section above. After one week, cells attached on the glass slides were washed with phosphate-buffered saline (PBS, 137 mM NaCl, 1.45 mM NaH_2_PO_4_.H2O, 20.5 mM Na_2_HPO_4_, pH 7.2) and fixed for 30 min at room temperature (RT) in 4% paraformaldehyde. After three washes with PBS + 0.01%Triton X-100 (PBST), cells were incubated at 4 °C overnight with anti-rabbit antibody (diluted 1:1000 in PBST) against *D. melanogaster* TRP-like neuropeptide, natalisin 4 (DromeNTL4) [50]. Then, the cells were washed three times for 5 min with PBST and incubated in dark conditions for 3 h at RT with goat anti-rabbit Alexa 488 conjugated secondary antibody (Life technologies) diluted at 1:1000. After three PBST washes, the cells were mounted in antifade media containing DAPI for nuclei staining (ProLong™ Diamond Antifade Mountant with DAPI, Thermo Fisher). The immunoreaction was observed under a Leica DMi8 confocal microscopy. The images were assembled in Adobe Photoshop CS4 (Adobe, Mountain View, CA, USA).

### 4.7. In Situ Hybridization

Validated ISH protocol previously developed by Šimo et al. [45,65] for tick synganglia was used. Briefly, a digoxygenin (DIG) probe synthesis kit (Roche Diagnostic, Germany) was used to synthesize a single-stranded DIG-labeled DNA probe for *trp* (662 bp). The *I. ricinus trp* ORF insert in the pGM-T Easy plasmid was used as a template (see the section Gene cloning and sequence analyses in Materials and Methods). Asymmetric PCR using either reverse (5′-CACGGCTTGGGGAATCTTCT-3′) or forward (5′-AGTGATAAGCAAACCCGGTG-3′) primer was performed to generate respective antisense and sense probes. Synthesized DIG-labeled probes were gel-purified and stored at –20 °C. Synganglia of unfed *I. scapularis* females were dissected in cold PBS and fixed with 4% paraformaldehyde for 2 h at RT. After cell membrane permeabilization with Proteinase K (New England BioLabs), synganglia were incubated with single-stranded DIG-DNA probes for 27 h at 48 °C. Then, specimens were incubated with mouse anti-digoxygenin/AP (Alkaline phosphatase; Roche Diagnostics, Germany) overnight at 4 °C. The reaction with hybridized DIG probes was developed by the addition of substrate/chromogen ready-to-use NBT-BCIP tablets (Roche Diagnostics, Germany). Finally, samples were incubated 5 min in 50% glycerol and subsequently mounted into 100% glycerol and observed by light microscopy (Olympus BX53). Images were assembled and enhanced in Adobe Photoshop CS4.

### 4.8. Wholemount Immunohistochemistry of Ixodes Synganglion

We slightly modified the IHC protocols previously described by Šimo et al. [38,45,47]. Briefly, *Ixodes* synganglia were dissected from unfed adult females, fixed with 4% paraformaldehyde solution for 2 h at RT, and then washed with PBS + 0.5% Triton X-100 (PBST). Tissues were incubated for 3 days at 4 °C with polyclonal anti-rabbit antibody against *D. melanogaster* NTL4 diluted 1:1000 in PBST. After three washes in PBST, the specimens were incubated two days at 4 °C with a goat anti-rabbit Alexa 488 conjugated secondary antibody (Life technologies) diluted at 1:1000. Samples were mounted in Prolong Antifade Diamond Mountant containing DAPI (Life Technologies) and analyzed by a Leica DMi8 confocal microscopy. Image assemblage was performed in Adobe Photoshop CS4. For neuronal cells of tick synganglia, we used nomenclatures as per Šimo et al. [38].

## Figures and Tables

**Figure 1 pathogens-10-00070-f001:**
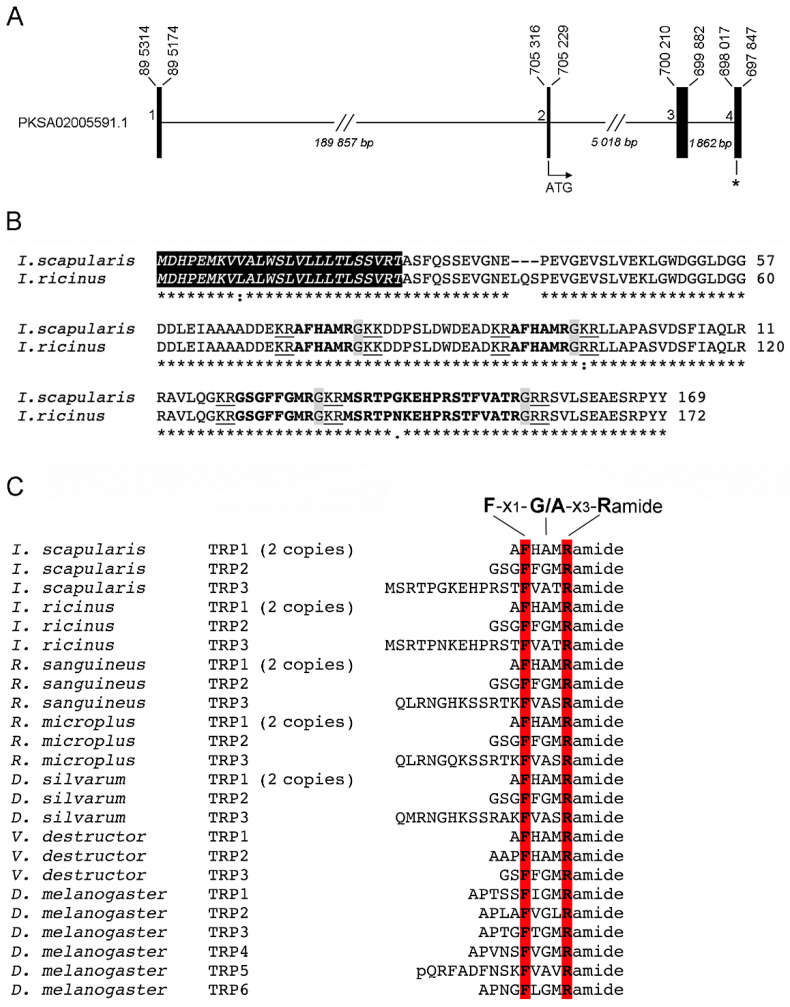
Genomic organization and open reading frame of *Ixodes* tachykinin-related peptide (TRP). (**A**) Exon/intron organization of *I. scapularis trp* in the ISE6 genome. Horizontal lines represent introns, while vertical lines represent exons (numbered). The ATG is the translation initiation signal, and the asterisk represent the stop codon. (**B**) Protein sequence alignment of *I. scapularis* (EL516783) and *I. ricinus* (MW082607) TRP prepropeptide sequences. The putative signal peptide at the amine terminus is in italics with a black background. Putative mature peptides are in bold fonts. Dibasic cleavage sites are underlined and the canonical amidation signal is in a gray background. Asterisk, single dot and double dot indicate identical, similar, and different amino acid residues, respectively. (**C**) An alignment showing the consensus of *I. scapularis, I. ricinus, R. sanguineus, R. microplus, D. silvarum, V. destructor,* and *D. melanogaster* mature TRP neuropeptides. The F and R amino acids residues of TRP F-x_1_-G/A-x_3_-Ramide motif are highlighted in a red background. *R. sanguineus, R. microplus,* and *D. silvarum* TRPs were predicted from GenBank Accession no. XP_037516719.1, XP_037279114.1, XP_037573036.1 respectively. *V. destructor* TRPs were extracted from Jiang et al. [49] and *D. melanogaster* TRPs were predicted from GenBank Accession no. CG14734.

**Figure 2 pathogens-10-00070-f002:**
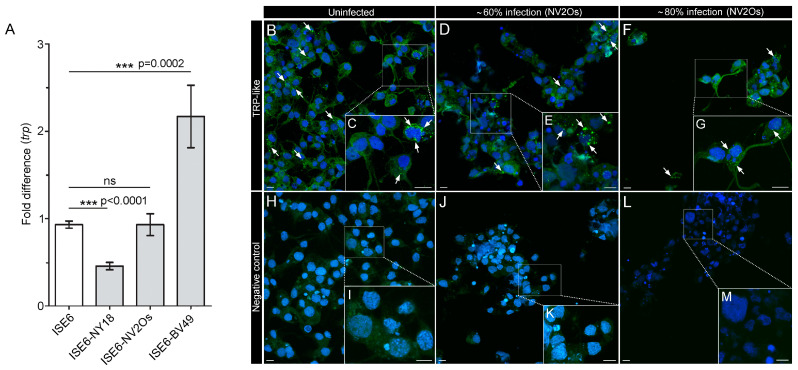
Expression of ISE6 TRP in response to *A. phagocytophilum* infection. (**A**) Fold changes in the transcript levels of TRP in the ISE6 cells (y-axis) infected 80% with three different strains (NY18, NV2Os, and BV49) of *A. phagocytophilum* (x-axis). The data from three distinct biological replicates were normalized using the *rps4* transcript. The asterisk (*) indicates the comparison of the standard mean error to the uninfected ISE6 value using a one-way Student’s *t*-test. (**B**–**G**) Immunocytochemistry (ICC) analyses of *I. scapularis* TRP in uninfected and NV2Os-infected ISE6 cells. (**B**,**C**) TRP-like IR (green, arrows) in uninfected cells. Inset in B is magnified in C. The TRP-like IR (green; arrows) for ≈60% (**D**,**E**) and ≈80% (**F**,**G**) infection of the cells with *A. phagocytophilum* NV2Os strain. Insets in (**D**) and (**F**) are magnified in (**E**) and (**G**), respectively. (**H**–**M**) Negative control staining (i.e., only the secondary antibody was used) for uninfected as well as ≈60% and ≈80% infected cells. Insets in (**H**–**L**) are magnified in (**I**–**M**,) respectively. Blue labeling are the nuclei 4′,6-diamidino-2-phenylindole (DAPI) staining. Scale bars are 10 μm.

**Figure 3 pathogens-10-00070-f003:**
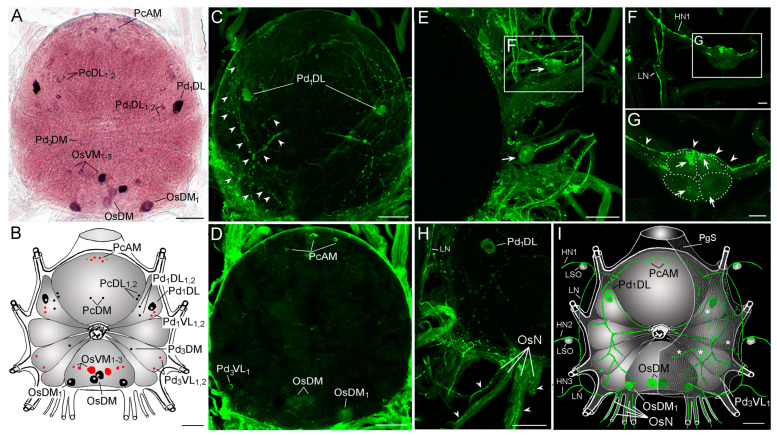
Wholemount *trp* in situ hybridization (ISH) and TRP immunohistochemistry (IHC) in synganglion from *I. scapularis* unfed female. (**A**) ISH staining with TRP antisense probe and schematic drawing in (**B**). In the schema (**B**), ISH-stained dorsal neurons are colored in black and ventral neurons are colored in red. (**C**) TRP-like IR (green) in *I. scapularis* synganglion. Arrowheads show the axonal projections originating from Pd_1_DL neurons. (**D**) Different Z-stack positions of the synganglion highlighting the TRP-like IR (green) neurons. (**E**) Lateral side of the synganglion highlighting the TRP-like IR (green) lateral nerve (LN) and associated lateral segmental organs (arrows). (**F**–**G**) Magnified view of the lateral segmental organs (LSO). Arrows in G show the TRP-like IR in cells of LSO, while arrowheads show the TRP-like axon within the hemal nerve 1 (HN1), running along the LSO. Dotted lines in G show the boundaries of the different LSO cells. (**H**) Lateral posterior part of the synganglion with associated nerves. Arrowheads show the TRP-like IR axons in four pairs of opisthosomal nerves (OsN), (**I**) Schematic drawing of *I. scapularis* synganglion summarizing all detected TRP-like IR (green) including neurons and axonal projections. Hemal nerves 1-3 (HN1-3), periganglionic sheath (PgS). Asterisks indicate TRP-like IR in axon terminals on dorsal surface of PgS. Scale bar for A–E is 50 μm, for F, G is 10 μm and for H, I is 50 μm.

**Table 1 pathogens-10-00070-t001:** Neuropeptide genes identified in ISE6 genome sequence and transcript databases. Note that Sequence Read Archive (SRA) corresponds to the Bioproject PRJNA239331 (containing 33 experimental datasets) for Illumina transcript reads. ^1^ Possible allelic forms of two scaffolds. ^2^ The gene likely spans multiple scaffolds (and multiple predictions); * Incorrect, partially predicted transcript (see Appendix A); ND—not detected. Note that XM, XR, and AXL predicted transcripts are from NCBI databases of ISE6 cell and were not detected in VectorBase ISE6 datasets. Nomenclature of the neuropeptides was used according to Coast and Schooley (2011) [51].

Neuropeptide Name	Computed Annotation	Scaffold	mRNA (SRA)
Adipokinetic hormone/corazonin-related peptide, (ACP)	XR_003917229	PKSA02006111.1	✓
Achatin-like (GFGE)	XM_029988349	PKSA02000030.1	✓
Allatostatin CC	ISCI001408	PKSA02005862.1	✓
Allatotropin	ISCI017791	PKSA02000866.1	✓
Arginine-vasopressin-like peptide (Inotocin)	XM_029985633 ^**1**^	PKSA02000317.1; PKSA02014421.1	✓
Bursicon alpha	ISCI004617	PKSA02001242.1	✓
Bursicon beta	ISCI004618	PKSA02001242.1	✓
Calcitonin-like diuretic hormone 34a	ISCI020490 ^**1**^	PKSA02012946.1; PKSA02003019.1	✓
Calcitonin-like diuretic hormone 34b	ISCI009341	PKSA02005071.1	✓
CCHamide-1	ISCI013057	PKSA02001506.1	✓
Corticotropin-releasing factor-related diuretic hormone	ISCI007845 ^**1**^	PKSA02003125.1; PKSA02006996.1	✓
Corazonin	ISCI014429	PKSA02006111.1	✓
Crustacean cardioactive peptide	ISCI010619	PKSA02006111.1	✓
Crustacean hyperglycaemic hormone/related ion transport peptide (CHH/ITP) isoform a	ISCI023228 ^**1**^	PKSA02003886.1; PKSA02006116.1	✓
Crustacean hyperglycaemic hormone/related ion transport peptide (CHH/ITP) isoform b	XM_029989718 ^**1**^	PKSA02003886.1; PKSA02006116.1	✓
Eclosion hormone	ISCI001941	PKSA02005732.1	✓
EFLamide	ISCW014582 ^**1**^	PKSA02010407.1; PKSA02009519.1	✓
Elevenin	AXL48134.1 ^**1,2**^	PKSA02008257.1; PKSA02016953.1 PKSA02007065.1; PKSA02009048.1	✓
FGLa-related allatostatin (Allatostatin A)	ISCI022939 ^**1**^	PKSA02013181.1; PKSA02004535.1	✓
Glycoprotein A2	XM_029991759 ^**1**^	PKSA02013868.1; PKSA02004257.1	✓
Glycoprotein B5	ISCI010926 ^**1**^	PKSA02004257.1; PKSA02012429.1	✓
Insulin-related peptide 1	ISCI002549 ^**1**^	PKSA02002782.1; PKSA02000328.1	✓
Insulin-related peptide 2	ISCW020331	PKSA02008623.1; PKSA02013879.1	✓
PISCF-related allatostatin (Allatostatin C)	ISCI001803	PKSA02005862.1	✓
Myoinhibitory peptide (Allatostatin B)	ISCI017595 ^**1**^	PKSA02004180.1; PKSA02003484.1	✓
Kinin	ISCI024200	PKSA02005591.1	✓
Natalisin	ISCI021632	PKSA02002188.1	ND
Neuroparsin	XM_0299932151	PKSA02004554.1	✓
Orcokinin	ISCI010518 64.1	PKSA02003264.1	✓
Proctolin	ISCI005701 ^**1,2**^	PKSA02003526.1; PKSA02018840.1 PKSA02018839.1	✓
Prothoracicotropic hormone-like (Trunk)	ISCI001809	PKSA02005517.1	✓
Pyrokinin	ISCI019582	PKSA02005732.1	✓
RYamide	ISCI005825	PKSA02000030.1	✓
SIFamide	ISCI022950	PKSA02005457.1	✓
Short neuropeptide F	ISCI007409	PKSA02002025.1	✓
Sulfakinin	XM_029979447	PKSA02001743.1	✓
**Tachykinin related peptide (TRP)**	** ISCI008383 * **	** PKSA02005591.1 **	**✓**
Trissin	ISCI011258	PKSA02005519.1	✓

## Data Availability

Not applicable.

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
