# Peer review of "Enlisting the Ixodes scapularis Embryonic ISE6 Cell Line to Investigate the Neuronal Basis of Tick—Pathogen Interactions"

_pathogens, 2021, doi:10.3390/pathogens10010070_

Round 1

Reviewer 1 Report

The manuscript PATHOGENS-1059987 entitled, ‘Enlisting the Ixodes scapularis Embryonic ISE6 Cell 2 Line to Investigate the Neuronal Basis of Tick-3 Pathogen Interactions’ describes on one side the existence of neuropeptide transcripts in the neuronal-like Ixodes scapularis embryonic cell line (ISE6) and presented their neuropeptidergic features, which resulted in the confirmation of the expression of a complete Ixodes neuropeptidome in ISE6 cells. On the other side, the authors described the neuropeptide tachykinin-related peptide (TRP) of ISE6 cells transcripts differentially depending on the infection strain of the tick-borne pathogen, Anaplasma phagocytophilum.

Congratulations on a very well written article with valuable data on an interesting topic. I found the methods used to be rigorous, with logical, and sequential steps. Moreover, I found the manuscript, figures, tables and supplementary materials to be clearly presented.

Author Response

We are gland that reviewer found the manuscript scientifically sounds and did not  mentioned any points to be addressed.

Reviewer 2 Report

The experimental design and the resulting data interpretation are scientifically sound. I have no major concern but only minor comments.

ISE6 cell line consists of several cells with different features and morphology. Did the author observe any difference in abundance of TRP between different cell types?

Figure 1B-C can be improved by including TRP of Rhipicephalus appendiculatus.

Author Response

Comment 1: ISE6 cell line consists of several cells with different features and morphology. Did the author observe any difference in abundance of TRP between different cell types?

Answer: We did not observed any differences in the TRP among different cell types therefore we noted that:

...the majority of uninfected ISE6 cells were positive to TRP.

Following the reviewer comment we modify the sentence to highlight the predominat type of cells: 

Although in predominant cell subpopulation, processes such as axon-like filopodia....

Comment 2: Figure 1B-C can be improved by including TRP of Rhipicephalus appendiculatus. 

Answer: Currently there is not R. appendiculatus TRP sequence available yet. But we included TRP mature peptide sequences of other three tick species in to the figure 1 C to improve it as suggested.